# Expression regulation of genes is linked to their CpG density distributions around transcription start sites

Hao Tian[1],* , Yueying He[1],*, Yue Xue[1],*, Yi Qin Gao[1,2,3]

The CpG dinucleotide and its methylation behaviors play vital roles in gene regulation. Previous studies have divided genes into several categories based on the CpG intensity around transcription starting sites and found that housekeeping genes tend to possess high CpG density, whereas tissue-specific genes are generally characterized by low CpG density. In this study, we investigated how the CpG density distribution of a gene affects its transcription and regulation pattern. Based on the CpG density distribution around transcription starting site, by means of a semi-supervised neural network we designed, which took data augmentation into account, we divided the human genes into three categories, and genes within each cluster shared similar CpG density distribution. Not only sequence properties, these different clusters exhibited distinctly different structural features, regulatory mechanisms, correlation patterns between the expression level and CpG/TpG density, and expression and epigenetic mark variations during tumorigenesis. For instance, the activation of cluster 3 genes relies more on 3D genome reorganization, compared with cluster 1 and 2 genes, whereas cluster 2 genes showed the strongest correlation between gene expression and H3K27me3. Genes exhibiting uncoupled correlation between gene regulation and histone modifications are mainly in cluster 3. These results emphasized that the usage of epigenetic marks in gene regulation is partially rooted in the sequence property of genes such as their CpG density distribution and explained to some extent why the relation between epigenetic marks and gene expression is controversial.

## Introduction

It is well known that the distribution of CpG dinucleotide is uneven along the genome sequence (Antequera, 2003). For instance, CpGs could accumulate to form the CpG island (CGI) (Antequera, 2003; Deaton & Bird, 2011). The CpG density around the transcription start site (TSS) is generally higher than its surrounding sequence given that most of the gene promoters are associated with CGI (Deaton & Bird, 2011; Vavouri & Lehner, 2012). Based on the normalized CpG density/GC content of promoter regions, genes have been divided into two or three categories, including HCP (high-CpG promoter), LCP (low-CpG promoter), and ICP (promoters with intermediate CpG contents) (Saxonov et al, 2006; Weber et al, 2007; Hartung et al, 2012; Couldrey et al, 2014; Yang et al, 2014). HCP genes tend to be housekeeping genes, whereas LCP genes are more likely to be tissue specific (Saxonov et al, 2006; Yang et al, 2014). These classifications were normally performed based on the average CpG intensity. As the distributions of CpG can also vary greatly within genes, it is interesting to explore whether the distribution in and around a gene affects its regulatory mechanisms (such as the deposition of epigenetic marks) and function.

The methylation of the cytosine of CpG represents a very important epigenetic mark which appears to be also dependent on the density distribution of CpG along the genome. The methylation level of CGI is usually low, especially in the promoter regions of highly expressed genes (Deaton & Bird, 2011; Vavouri & Lehner, 2012; Aoto et al, 2020). Besides, polycomb repressive complex 2 (PRC2), which is thought to participate in the methylation of lysine on H3, tends to bind to CpG-dense regions under the help of polycomb-like proteins (van Kruijsbergen et al, 2015; Li et al, 2017). The mutual exclusion between CGI methylation and trimethylation of H3K27 was observed in both human and mice cells (Bogdanovic et al, 2011; Lynch et al, 2012). H3K27me3-marked DNA methylation canyons can form long-range chromatin interactions, which was associated to specific gene repression (Zhang et al, 2020). In general, epigenetic marks, including DNA methylation; active histone marks H3K4me3, H3K36me3, and H3K27ac; and repressive histone marks H3K27me3 and H3K9me3, are related to gene activation or repression (Santos-Rosa et al, 2002; Heintzman et al, 2007; Benayoun et al, 2014; Jang et al, 2017; Ninova et al, 2019). Notably, at the same time, many studies also showed the decoupling of epigenetic marks from gene regulation (Murray et al, 2019 Preprint; Borsari et al, 2020 Preprint). It is thus intriguing to investigate the possible reasons behind such inconsistency, in particular, whether the DNA sequence plays a role in the usage of different epigenetic regulation.

[1]Beijing National Laboratory for Molecular Sciences, College of Chemistry and Molecular Engineering, Peking University, Beijing, China   [2]Biomedical Pioneering Innovation Center (BIOPIC), Peking University, Beijing, China   [3]Beijing Advanced Innovation Center for Genomics (ICG), Peking University, Beijing, China

Correspondence: gaoyq@pku.edu.cn
*Hao Tian, Yueying He, and Yue Xue contributed equally to this work.

To address the questions described above, we performed in this paper gene classification based on the CpG density distribution around TSS using a semi-supervised neural network. The analyses revealed that, in general, human genes can be divided into three categories. Cluster 1 genes possess a high and sharp CpG density peak around TSS. Cluster 2 genes harbor a lower but broader CpG density peak, compared with cluster 1 genes. In contrast, cluster 3 genes are characterized by low CpG densities around TSS. Not only the sequence property, the patterns of nucleosome occupancy, and transcription factor (TF) binding, the correlation between the gene expression level and CpG/TpG density, the regulatory mechanisms, and the epigenetic mark and expression variations during tumorigenesis are also distinctly different among the three gene clusters. Together, our results emphasized the importance of taking the genetic sequence properties into account for understanding the gene regulatory mechanisms.

## Results

### Gene classification based on CpG distribution

Many studies have revealed the distinctly different CpG distributions around TSS between housekeeping genes and tissue-specific genes (Saxonov et al, 2006; Roider et al, 2009; Yang et al, 2014). Here, we first used the Recurrent Neural Network (see the Materials and Methods section) to distinguish most of the housekeeping genes from tissue-specific genes (Fig 1A). Briefly, we downloaded the human promoter CAGE data from the FANTOM5 project (https://fantom.gsc.riken.jp/5/datafiles/latest/extra/CAGE_peaks/) and regarded the midpoint of CAGE peaks labeled with "p1@" as gene TSS. For each gene, the corresponding CpG density distribution was calculated based on the DNA sequences of 16 kb (8 kb upstream to 8 kb downstream of TSS) using the 40-bp nonoverlapping window, thus generating a 1 × 400 vector. The network was first trained (see the Materials and Methods section) using the CpG density distributions of a proportion of housekeeping genes (downloaded from UCSC database) and tissue-specific genes (supervised loss) (Tian et al, 2020) and the unlabeled genes (unsupervised loss, Fig 1A). The AUC value of the testing set is 0.947.

Next, we applied this well-trained model to the entire gene set, each gene was therefore conferred by a "CG likelihood," the value of which indicates the confidence that one gene could be regarded as a tissue-specific gene. Similar to previous studies (Weber et al, 2007), the distribution of CG likelihood roughly consisted of three parts (Fig 1B), indicating that the human genes could be divided into three clusters, and we performed such a classification by means of a Gaussian mixture model. Genes in clusters 1 (gene number is 12361) and 3 (gene number is 8396) possess the lowest and highest CG likelihood and are characterized by highest (and sharp) and lowest CpG density distribution around TSS, respectively, whereas cluster 2 genes (gene number is 3752) have broad CpG peaks around TSS featured by moderate intensity (Fig 1C). Accordingly, almost all (3,123 of 3,447) housekeeping genes are in cluster 1, and in contrast, the most of (1,292 of 1,821) tissue-specific genes belong to cluster 3. Genes in clusters 1 and 3 harbor the lowest and highest tissue specificities (higher value of this parameter indicates the

corresponding genes are specifically and highly expressed in fewer tissues [Tian et al, 2020], see the Materials and Methods section), respectively (Fig 1D), accordant with previous studies revealing the negative correlation between TSS CpG density and the tissue specificity score (Yang et al, 2014). Notably, although tissue-specific genes tend to possess low CpG densities around TSS, most tissue-specific (ts)-TFs are relatively CpG-rich, given that 108 of 171 ts-TFs are in cluster 2. We also analyzed the distribution of tissue-specific genes pertinent to certain tissues (e.g., liver-specific genes) among three clusters. Interestingly, brain-specific genes tend to reside in clusters 1 and 2, whereas liver-, spleen-, and whole blood-specific genes are more likely to belong to cluster 3 (Fig S1A).

Akin to CpG density distribution, the nucleosome occupancy patterns are also found to be significantly different among the three clusters. In general, as for cluster 1 genes, the promoter regions of which reside in an open environment, and the arrangement of nucleosome downstream of TSS is regular (Fig 1E). Such a feature can also be observed for cluster 2 genes but to a much lesser extent (Fig 1E). By contrast, cluster 3 genes are generally located in a compact and nucleosome-occupied environment, especially for promoter regions (Fig 1E). A recent study has uncovered the binding sites (ChIP-seq peaks) of 208 chromatin-associated proteins (CAPs), including 171 TFs and 37 transcriptional cofactors and chromatin regulator proteins in HepG2 cells (Partridge et al, 2020). We found that the binding sites of most of the CAPs, for example, ERF, ELF3, CHD2, and MAZ, tend to reside in a specific region for cluster 1 genes, whereas the binding patterns become more "dispersed" for clusters 2 and 3 genes (Figs 1F and S1B).

The housekeeping genes and tissue-specific genes in cluster 1 (named c1-HKGs and c1-TSGs, respectively, although the number of the latter is relatively low: 3,123 versus 221) possess very similar CpG distribution patterns (Fig 1G), and they both locate in an open environment (Fig 1E). To understand the factors contributing to their different transcriptional activities, we examined their CAP binding patterns. Because in contrast to c1-HKGs, many c1-TSGs are not expressed in HepG2, we chose as examples the highly expressed c1-TSGs (named h-c1-TSGs), the expression level of which exceeds the 75th percentile of all genes' expression level and is comparable to c1-HKGs (*P*-value > 0.05). One can see from Fig 1H that CAPs are inclined to bind to the promoter regions of c1-HKGs, hinting that the sequence motif of c1-HKGs promoter regions is more likely to recruit TFs. We also compared the CAP binding for highly expressed c1-TSGs and c3-TSGs and found that CAPs with higher tissue specificities have a higher tendency to bind to c3-TSGs (Fig S1C), which means that these CAPs are prone to regulate tissue-specific genes with low but not high promoter CpG densities. Aside from sequence features, we also examined the 3D chromatin structure properties and found that the insulation score (one parameter used to assess the possibility that the locus locates in TAD boundary, see the Materials and Methods section) of c1-HKGs is significantly higher than c1-TSGs (Fig S1D). This result shows that compared with c1-TSGs, c1-HKGs are more likely to reside at TAD boundaries, accordant with previous findings that HKGs are enriched near TAD boundaries (Dixon et al, 2012). Together, these results revealed that although the CpG distributions around the TSS are similar, c1-HKGs and c1-TSGs display distinctly different

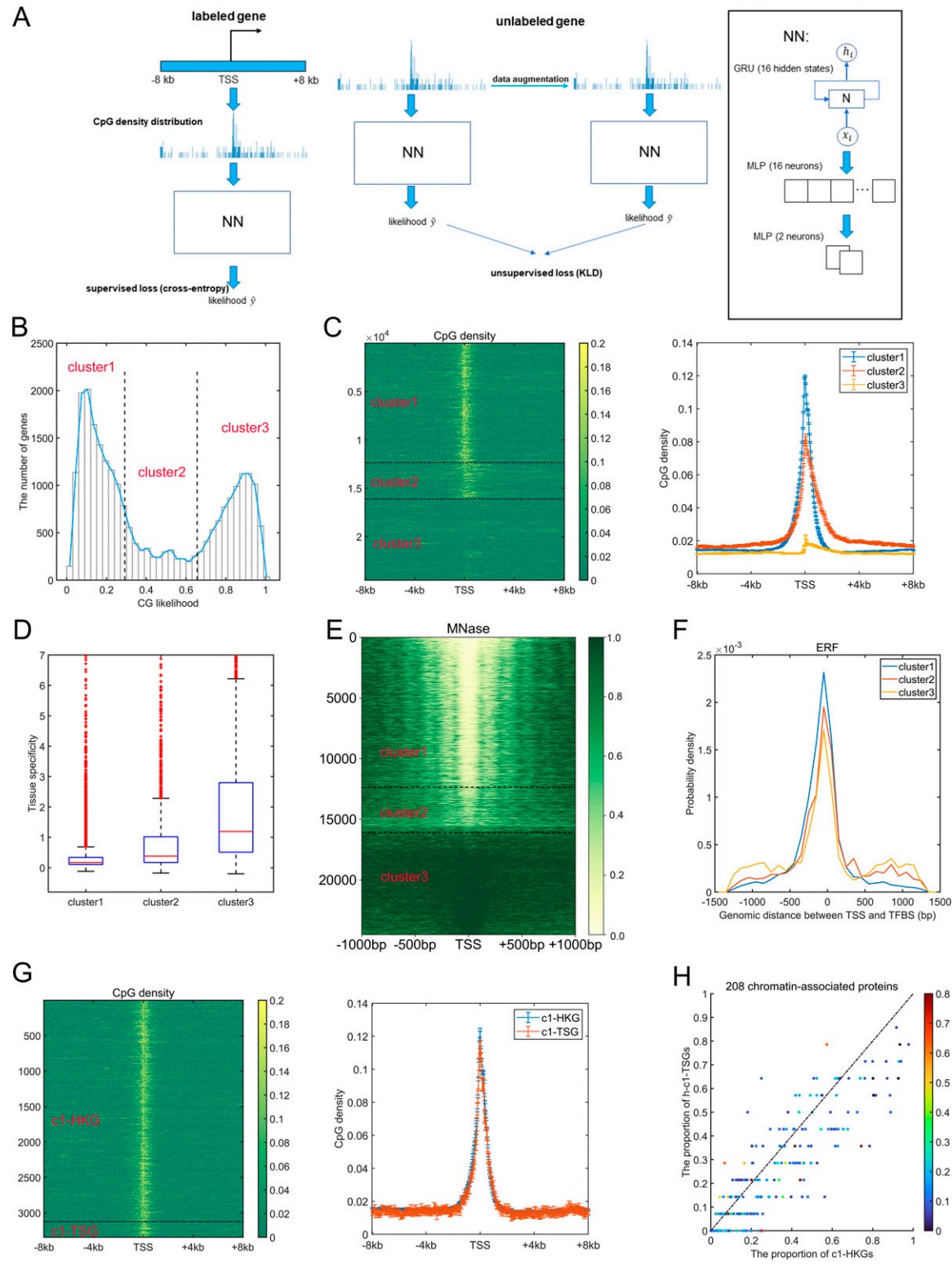

**Figure 1.   Gene classification based on CpG distribution.**
**(A)** Overview of the network-training process. **(B)** The distribution of CG likelihood of human genes. **(C)** The CpG density distribution of genes belonging to different clusters (left, heatmap; right, average behaviors). **(D)** The tissue specificity of genes of three clusters. Based on the definition of tissue specificity (see the Materials and Methods section), for each gene, its maximum tissue specificity among 37 tissues (testis was not considered because it contains too many tissue-specific genes) was extracted and drawn here. $P$-values = $3.12 \times 10^{-69}$ (cluster 1 versus cluster 2), $6.37 \times 10^{-276}$ (cluster 1 versus cluster 3), and $4.28 \times 10^{-106}$ (cluster 2 versus cluster 3) by Welch's unequal variance $t$ test. **(E)** The nucleosome occupancy patterns (measured by MNase-seq) of genes of three clusters in GM12878. **(F)** The distribution of genomic distance between transcription factor (TF)–binding sites and gene transcription starting site. Positive and negative values indicate that TFs bind to the regions downstream and

sequence and structure traits. The sequence properties for one gene to be a HKG include not only the high CpG density within the promoter region, which not only permits the open chromatin structure (Fig 1E) but also the specific sequence motif which can effectively recruit different kinds of TFs for transcription.

We note here that there exist many studies (Saxonov et al, 2006; Weber et al, 2007; Hartung et al, 2012; Couldrey et al, 2014; Yang et al, 2014) dividing the gene promoters into two or three categories, for example, HCPs (high-CpG promoters), LCP (low-CpG promoters), and ICP (promoters with intermediate CpG content). However, the criterion behind these classifications was based on the (normalized) promoter CpG/G+C content and did not consider explicitly the CpG density distribution upstream and downstream of TSS. Here, as an example, we first calculated the overlap between gene clusters identified here and by Weber et al (2007) and found that most HCP and LCP genes belong to cluster 1 and 3, respectively, and ICP is distributed among all three clusters (Fig S1E). As expected, almost all cluster 1 genes are associated with CGI, whereas only a minority of cluster 3 genes is associated with CGI (Fig S1F). Such an association was also observed when we compared the genes of three clusters with nonmethylated islands identified by Bio-CAP (Long et al, 2013) (Fig S1G). Furthermore, we compared the CpG density distribution around TSS and tissue specificity between clusters identified by RNN and Weber et al (2007). Minor difference existed within clusters identified by RNN (Fig S1H and I), indicating that our classification does provide opportunities to investigate how sequence property variation affects gene regulatory mechanisms. In fact, genes within each cluster (identified in this work) exhibit particular regulatory mechanisms that are significantly different from each other, which will be discussed below.

### The relation between sequence property and gene expression

Earlier studies revealed a positive correlation between CpG density around TSS and gene expression in vertebrates (Cheng et al, 2012; Yang et al, 2014). This result is not surprising because housekeeping genes are characterized by high CpG density in promoter regions and are normally highly expressed. Here, we asked whether such a positive correlation between expression and CpG density still holds in each individual cluster. We calculated the Spearman correlation coefficient between the gene expression level and CpG density of each nonoverlapping 40-bp window among genes belonging to the same cluster (see the Materials and Methods section) and found that CpG densities around TSS are in general positively correlated with gene expression, regardless of the cluster under study (Figs 2A, S2, and S3). Furthermore, we noticed that this correlation is higher for clusters 2 and 3 genes than for cluster 1 genes in most samples we examined (including early embryonic cells, somatic cells, tumor cells and its corresponding paracancerous cells, Figs 2A, S2, and S3). In a few samples (e.g., early embryonic cells, hES, LIHC, and BLCA), we did not observe the prominent correlation peak near the TSS (Figs 2A and S3) for cluster 1 genes. The correlation level

downstream of TSS (TSS—approximately +4 kb) decays much slower for cluster 3 and is thus more positive than clusters 1 and 2 genes (Figs 2A and S2). These results thus show that aside from sequence properties, the correlation patterns derived from different gene clusters are also significantly different (therefore, the correlation pattern may be influenced intrinsically by sequence properties).

An earlier study revealed that the expression level was negatively correlated with TpG density near TSS of pig genes (Schachtschneider et al, 2015). We found here that in different human cells, these three clusters exhibit an overall negative correlation between expression and the TpG density near TSS, but this correlation level for clusters 2 and 3 genes is generally more negative than cluster 1 genes (Figs 2B and S4–S6). Intriguingly, as for clusters 1 and 2, the correlation between expression and TpG density 0 ~ 40 bp downstream of TSS is positive and thus noticeably different from its surrounding sequences (Figs 2B, S4, and S5). In the following, we show that this region also distinguishes itself by unique histone modification marks.

### Distinct regulatory mechanisms among three gene clusters

Because the three gene clusters have distinct sequence features and epigenetic mark usage is expected to be affected by DNA sequence, we next investigated whether the corresponding epigenetic mechanisms also differ for the three classes of genes. We analyzed the RNA-seq, histone modification, and DNA methylation data of the liver, lung, ovary and sigmoid colon. Similar results were obtained for each set of data. Taking the liver as an example, only a small proportion of cluster 1 genes are marked with repressive histone modifications (H3K27me3 and H3K9me3, Figs 3A and S7A), and as expected, these genes possess relatively low expression levels within cluster 1 (Figs 3A and S7A). Most cluster 1 genes are decorated with active histone marks (H3K4me3, H3K27ac, and H3K36me3) and possess low DNA methylation levels around TSS, consistent with their higher expression levels (Figs 3B and S7B–D). Intriguingly, one feature for repressed genes of cluster 2 is that their high H3K27me3 signals tend to be dispersed along the genome sequence upstream and downstream of TSS (Fig 3A). In contrast, the H3K27me3 distributions of cluster 1 repressed genes are limited close the TSS (Fig 3A). For cluster 3 genes, although a higher proportion of genes are repressed, compared with clusters 1 and 2, a dispersed and much weaker pattern of H3K27me3 is observed for repressed genes (Fig 3A), indicating that the repression of cluster 3 genes may be largely independent of the deposition of H3K27me3. Such results are accordant with previous studies, revealing that PRC2, playing a vital role in trimethylation of H3 on lysine K27, tends to bind to CG-rich regions (Mendenhall et al, 2010; van Kruijsbergen et al, 2015; Li et al, 2017).

To quantify the expression dependences on epigenetic modifications among genes with different DNA sequences, akin to the correlation between CpG density and gene expression level introduced above, for each cluster, we calculated the spearman

---

upstream of transcription starting site, respectively. *P*-value = 2.18 × 10$^{-9}$ for cluster 1 and 2, *P*-value = 0.04 for cluster 1 and 3. Welch's unequal variance *t* test. **(G)** The CpG density distribution of c1-HKG and c1-TSG (left, heatmap; right, average behaviors). **(H)** The proportion of c1-HKGs/h-c1-TSGs that bind to one certain TF. Each data point represents one certain TF, and the corresponding color represents its tissue specificity. *P*-value = 4.12 × 10$^{-11}$ by *t* test.

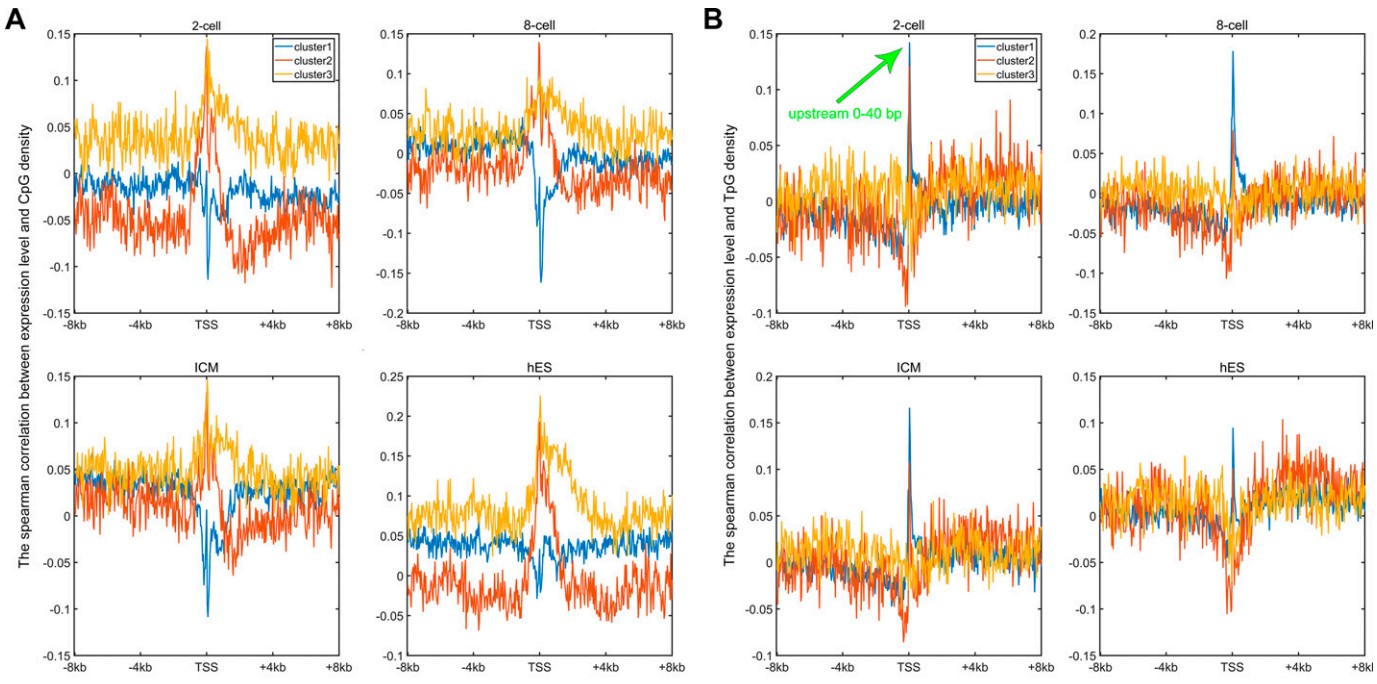

**Figure 2. The relation between sequence property and gene expression.**
**(A)** The Spearman correlation between gene expression and CpG density of each nonoverlapping 40-bp window (see the Materials and Methods section); the expression level of three gene clusters in these four cells could be seen in Fig S6. For each cell, *P*-value < 0.001 when two correlation curves were compared (*t* test). **(B)** The Spearman correlation between gene expression and TpG density of each nonoverlapping 40-bp window. For each cell, *P*-value < 0.001 when two correlation curves were compared (*t* test).

correlation between expression levels and epigenetic marks around the TSS for different genes (Figs 3C and S8). The correlation between the expression level and DNA methylation is negative near TSS and is positive both up- and downstream of TSS for all three types of genes (Fig 3C). Notably, among all three gene clusters, the expression level of cluster 2 genes is most strongly correlated to DNA methylation. As for repressive histone modifications (H3K27me3 and H3K9me3), the gene expression is negatively correlated with the modification level, especially for cluster 2 genes (Fig 3C). Interestingly, we found that the TF EZH2 (Partridge et al, 2020), which is thought to be involved in the methylation of H3K27, tends to bind to the promoter regions of cluster 2 genes (*P*-value = $6.67 \times 10^{-20}$ for cluster 1 and 2, *P*-value = $3.42 \times 10^{-76}$ for cluster 2 and 3, Fisher's exact test), indicating that the wide usage of repressive histone mark for the down-regulation of cluster 2 genes. For active histone modifications, as expected, the expression is positively correlated with H3K4me3 and H3K27ac signals around TSS and with H3K36me3 signal in gene body (Fig 3C). These correlation levels are generally lower for cluster 1 genes than for genes of the other two clusters (Fig 3C).

The above results revealed that different gene clusters exhibit distinct correlation patterns between the expression level and epigenetic marks, hinting the important role of DNA sequence feature in the aspect of epigenetics. In fact, as we introduced above, the H3K27me3 distribution of repressed genes belonging to different clusters does correlate with the corresponding CpG density distribution. In fact, cluster 3 genes tend not to be occupied by H3K27me3 for repression, resulting in the low correlation between

H3K27me3 and the expression level. In contrast, cluster 2 genes tend to recruit proteins responsible for trimethylation of H3K27 (such as polycomb group proteins) for repression because of their sequence characteristics (likely, the high and broad CpG density peak), leading to the higher correlation between expression and H3K27me3 signal. A recent study (Borsari et al, 2020 *Preprint*) investigated the variation of expression and nine histone modification signals of human coding genes along the transdifferentiation process, from pre-B cell to macrophage, and identified a gene cluster: genes within which are not marked by most kinds of histone modifications throughout the differentiation process but exhibit expression variation. Intriguingly, the majority (70%) of these genes are found here to belong to cluster 3, indicating that the expression level of genes characterized by low CpG density are more likely uncoupled from histone marks. To gain information on the regulatory mechanism of cluster 3 genes, we calculated the correlation between the compartment index (gene with higher value indicates it locates in a more compartment A environment, and compartments A and B are mainly corresponding to euchromatin and heterochromatin, respectively, see the Materials and Methods section) and gene expression level among different tissues and found that this correlation for cluster 3 genes appears to be higher than other two types of genes (Fig S7E). Such a result indicates that chromatin structure organization plays a more important role in cluster 3 gene regulation: these genes are intrinsically prone to silencing, and their activation are likely facilitated by a movement to a more compartment A–like environment, possibly under the help of specific TFs (Hnisz et al, 2017;

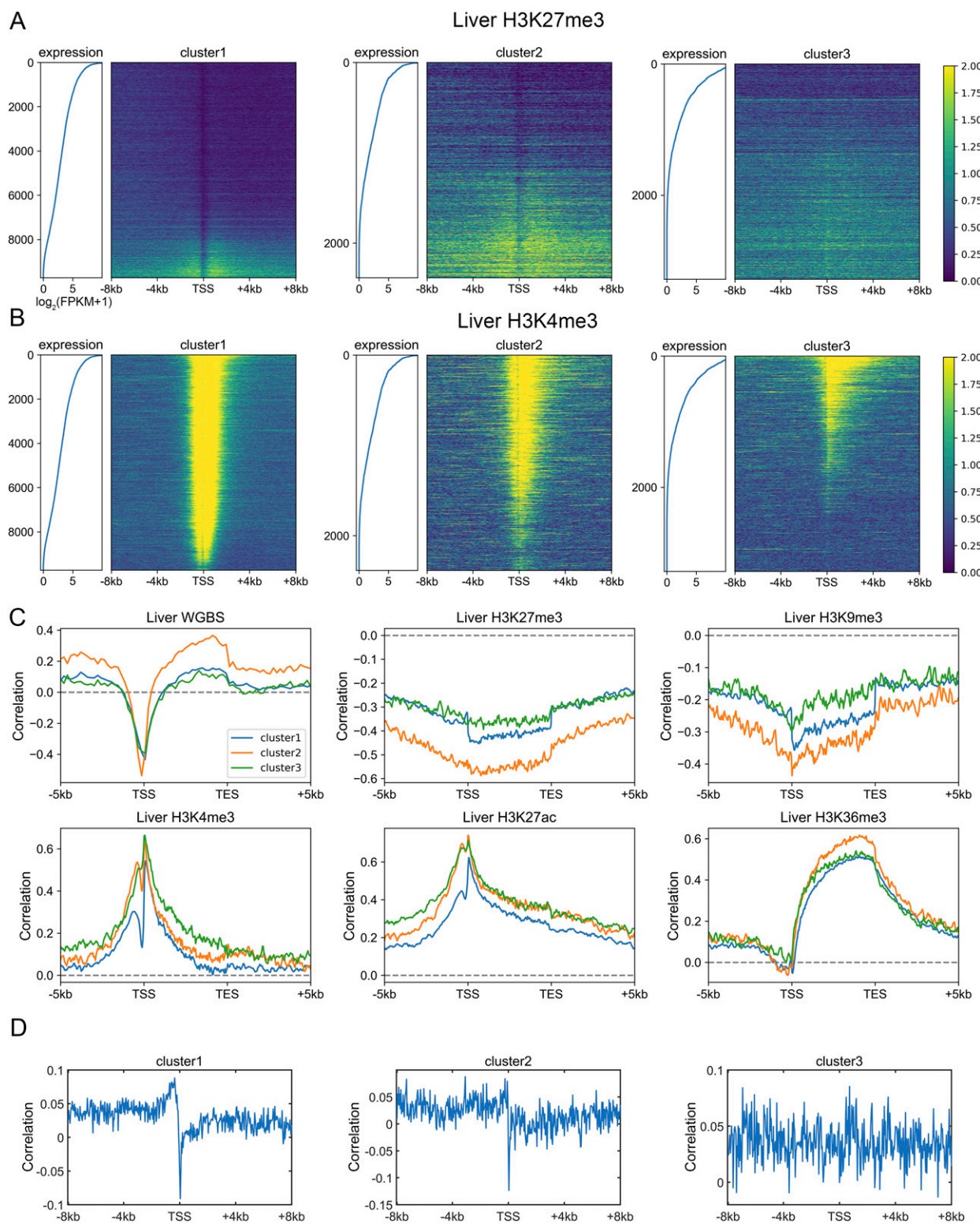

**Figure 3. Distinct regulatory mechanisms among three gene clusters.**
**(A, B)** The distribution of H3K27me3 (A) and H3K4me3 (B) among three gene clusters in the liver. Each heatmap was ranked based on the gene expression level. ChIP-seq signal represents the fold change (ChIP-seq counts relative to control). **(C, D)** The Spearman correlation between the gene expression level and epigenetic marks (C) and between TpG density and H3K27me3 (D) (see the Materials and Methods section). For Fig 3C, all $P$-values < $10^{-7}$ and for Fig 3D, all $P$-values < $10^{-4}$ ($t$ test).

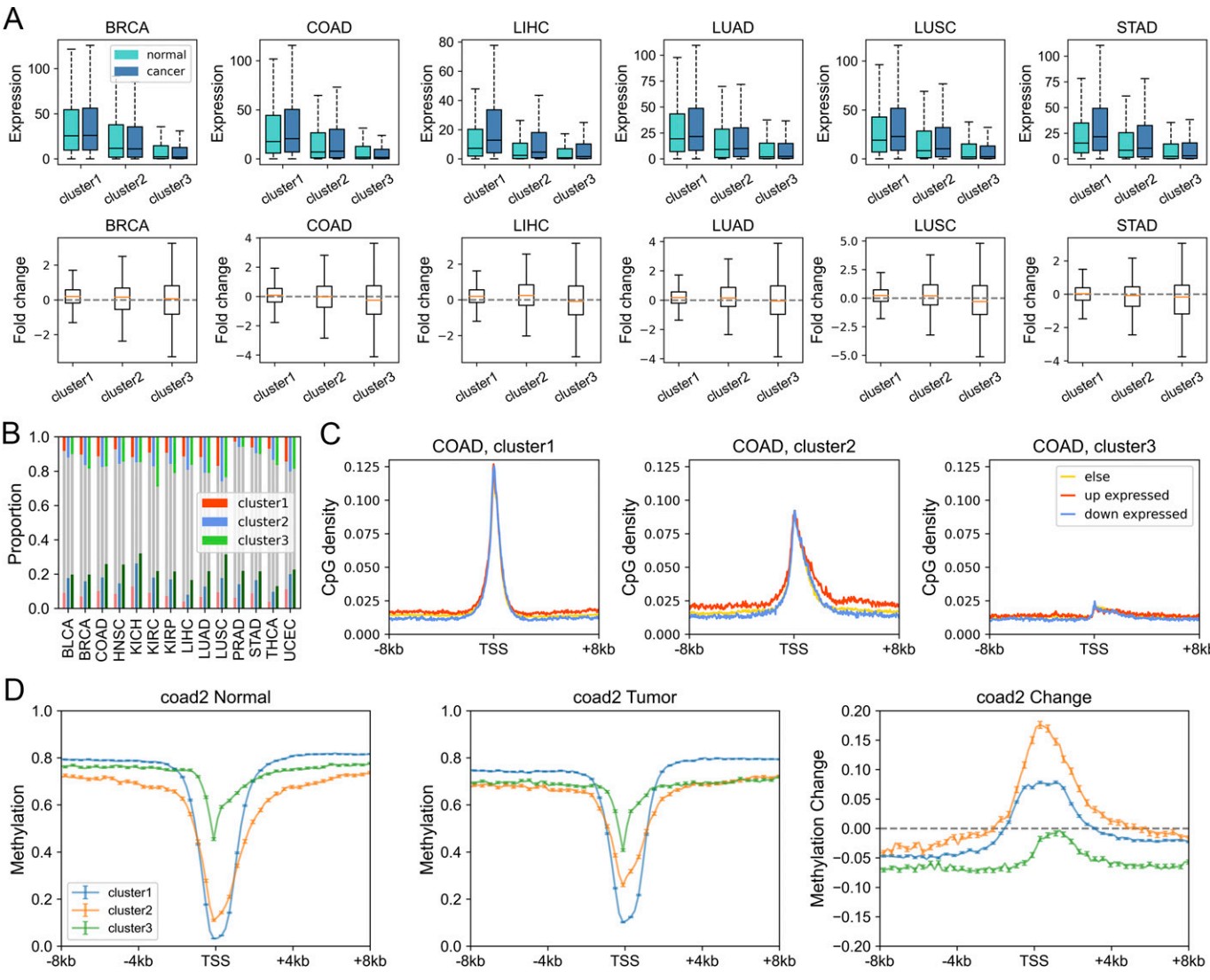

**Figure 4. Expression and epigenetic changes in carcinogenesis.**
**(A)** The expression level (TPM) of genes of different clusters in cancer and normal samples (upper) and the $\log_2$ (expression fold change) in carcinogenesis calculated by DESeq2 (down). **(B)** The proportion of DE genes (in carcinogenesis) in three clusters. Upper: the proportion of up-expressed genes, down: the proportion of down-expressed genes. **(C)** The CpG density distribution of up-expressed, down-expressed, and other genes. COAD (colon cancer) was used here for illustration. See Fig S12 for more instances. **(D)** The DNA methylation level of genes of different clusters in normal (left) and tumor (middle) cells and the methylation changes during carcinogenesis for three gene clusters (right). COAD was used for illustration. See Figs S14 and S15 for more instances.

Hnisz & Young, 2017; Boija et al, 2018; Kim & Shendure, 2019; Stadhouders et al, 2019; Tian et al, 2020).

As mentioned above, cluster 1 and 2 genes show a sharp positive peak in the correlation between the expression level and TpG density 0 ~ 40 bp downstream of TSS (Fig 2B). Consistently, we found a salient negative correlation between TpG density 0 ~ 40 bp downstream of TSS and repressive histone modifications (H3K27me3, Fig 3D).

## Expression and epigenetic changes in carcinogenesis

Given that gene dysregulation is a hallmark of cancer (Hanahan & Weinberg, 2011), we next investigated whether genes belonging to different clusters show different expression variations in carcinogenesis. In general, during tumorigenesis, housekeeping-like

cluster 1 genes tend to be up-regulated, whereas cluster 3 genes show an opposite tendency (Fig 4A, P-values [for cluster 1 and 3] and were less than 0.001 by Welch's unequal variance t test in all six cancer types; the volcano plots of each cluster could be found in Fig S9A). For instance, *SLC16A1* and *BSG*, belonging to cluster 1 and playing vital roles in energy metabolism (Halestrap & Price, 1999; Felmlee et al, 2020), are up-regulated in a variety of cancer types (Fig S9B) and could promote tumor growth and aggressiveness through the Warburg effect (Vaupel & Multhoff, 2021). This result is consistent with our previous finding that genes with high CpG densities tend to be up-regulated in cancer cells (Xue et al, 2022), indicating that the expression change in carcinogenesis is partially coupled with the CpG density. Nevertheless, Fig 4A shows that although cluster 1 genes are more likely to be up-regulated, the

changes tend to be small. By comparison, cluster 3 genes display a broad distribution of expression fold change (Fig 4A), suggesting that some genes are dramatically activated/repressed in tumorigenesis (besides, we also performed the shrinkage of logFC and the results remain the same [Fig S9C]). In addition, consistent with our previous findings (Xue et al, 2022), genes specifically and highly expressed in paracancerous tissue (mainly in cluster 3) are mostly down-regulated in cancer cells (Fig S10A), whereas complementary genes (i.e., genes specifically expressed in other tissues other than paracancerous) are mostly up-regulated (Fig S10B).

Next, we investigated the DNA sequence characteristics, functions, and epigenetic modifications of genes, the expression variations of which are large in carcinogenesis (hereafter referred to as DE genes, which are identified by DESeq2 [Love et al, 2014], see the Materials and Methods section). The proportions of DE genes in clusters 1 and 3 are lowest and highest, respectively, in all cancer types (Figs 4B and S11), consistent with the above results that expressions of cluster 1 genes change the least and those of class 3 genes the most in carcinogenesis. The CpG density up- and downstream of TSS for up-regulated genes are slightly higher than other genes almost for all types of cancers, especially for cluster 2 genes (Figs 4C and S12). GO function analyses show that up-regulated genes in cluster 1 are closely related to cell cycle, including nuclear division, DNA replication, and chromosome segregation, contributing to the uncontrolled growth of tumor (Fig S13). Down-expressed genes in clusters 1 and 3 are often related to the muscle system process (Fig S13). The DE genes in cluster 2 are found to be associated with development, including embryonic organ morphogenesis, cell fate commitment (Fig S13), which were suggested to relate to the dedifferentiation and invasion of cancer cells (Ma et al, 2010).

On the other hand, we found that genes enriched in H3K27me3 in paracancerous (normal) tissue, which is a standing-out property for developmental genes (Schuettengruber et al, 2017), tend to become dysregulated in cancer cells, especially for cluster 2 genes. To be specific, for colon cancer, compared with stably expressed genes (in carcinogenesis), up- and down-expressed genes have significantly higher H3K27me3 levels in their corresponding normal tissues (Fig S14A). At the same time, the TSS of these genes is also likely to be hypermethylated in carcinogenesis (Fig S14A). Traditionally, DNA hypermethylation around TSS is associated with the gene repression, and this hypermethylation-associated gene activation in cancer is possibly a result of their repression by a broad H3K27me3 signal around TSS in normal cells (Fig S14A). Not only the TSS but also the upstream and downstream of TSS of these up-expressed genes are hypermethylated in cancer cells (Fig S14A), with the latter being favorable for gene expression (Schachtschneider et al, 2015; Luo et al, 2018). In addition, the TSS of cluster 2 genes undergo the most significant hypermethylation (Figs 4D, S14B, and S15A–C) among the three types of genes, consistent with them being the most enriched in H3K27me3 in normal cells because many studies have revealed that most promoters that exhibit cancer-associated hypermethylation are linked to genes silenced by polycomb and H3K27me3 in corresponding normal tissues (Schlesinger et al, 2007; Ehrlich, 2019). For instance, the TSS of several developmental-related and cluster 2 genes, such as *GATA4*, *HOXD11*, and *HOXD12*, was hypermethylated in a variety of cancer types (Fig S15D). Finally,

the up- and downstream of TSS of cluster 3 genes tend to be more hypomethylated than other genes (Figs 4D, S14B, and S15). In fact, our previous work (Xue et al, 2022) found that loci with low CpG density are located in repressive and inaccessible environment (compartment B) and are more likely to be hypomethylated in cancer cells. Accordingly, cluster 3 genes are CpG-poor and mainly located in the repressive compartment B, in which hypomethylation is more likely to be found.

## Discussion

To investigate how much CpG density distribution of a gene affects its expression and regulation pattern, we performed in this study the gene classification using recurrent neural network. We then analyzed the biological functions and the regulatory mechanisms of genes with different sequence properties. Compared with previous methods, which classified genes based on CpG/C+G intensity of promoter regions (Saxonov et al, 2006; Weber et al, 2007), our classification considered the CpG distribution in a large region around the TSS and thereby included more comprehensive sequence features. The model yielded three gene clusters, each with considerably different sequence features. We found that three gene clusters are distinctly different in terms of expression, regulatory mechanisms, chromatin structural features, and TF-binding patterns. For instance, cluster 1 genes tend to reside intrinsically in an open chromatin environment and possess the highest expression level, and their activation relies weakly on specific genome reorganization. Benefiting from the unique chromatin environment (possible because of the unique sequence property), cluster 1 genes are more likely to stably express (almost all housekeeping genes are in cluster 1), thus contributing to the low correlation between expression level and CpG density. In contrast, the promoter regions of cluster 3 genes have low CpG densities and accessibility. Tissue-specific TFs (such as pioneer factors [Iwafuchi-Doi & Zaret, 2014], as discussed below) may play a vital role in the regulation of cluster 3 genes given that these factors could overcome the nucleosome barriers and tether the cluster 3 genes to a more active environment (e.g., through a phase separation mechanism) (Hnisz et al, 2017). Notably, our gene classification clearly identified a gene cluster, the promoter regions of genes within which are characterized by a high and broad CpG peak (cluster 2). The regulation of cluster 2 genes was found to be most strongly correlated with epigenetic modifications, especially for H3K27me3, among all three types of genes. Again, we compared the correlation patterns (between expression level and epigenetic marks) between clusters identified here and by Weber et al (2007) and found that cluster 2 exhibited a stronger correlation level between the expression level and H3K27me3 compared with ICP (Fig S16). Given that the association between epigenetic mark and gene expression is still controversial and different gene clusters exhibit different degrees of dependence on epigenetic marks, we speculate that the confusion may, at least in part, result from the diversity of CpG distribution around promoter regions of different genes.

Enhancers have been suggested to play vital roles in gene regulation, given that they provide accommodations for TFs and

could form spatial contact with promoters to activate specific genes (Panigrahi & O'Malley, 2021). In general, H3K27ac/H3K4me1/ATAC/DNase peaks were often regarded as candidate enhancers (Oka et al, 2017; Crispatzu et al, 2021; Ing-Simmons et al, 2021). Then the candidate enhancer nearest to genes was thought to regulate that gene. However, with the development of chromosome conformation capture technology, many studies have revealed that enhancer could skip its nearest gene and regulate genes far away from that enhancer (Krivega & Dean, 2012; Wang et al, 2021). Akin to this, our work revealed that the activation of cluster 3 genes (mainly residing in heterochromatin) relied more on movement toward compartment A, indicating that the enhancers regulating cluster 3 genes may be mainly in compartment A (akin to the phase separation mechanisms [Hnisz et al, 2017]), suggesting the longer genome distance between cluster 3 genes and its corresponding enhancer, compared with cluster 1 and 2 genes. As mentioned earlier, because the main purpose of this study was to investigate whether relatively simple sequential features of a gene are associated with their expression patterns in different cells and whether these sequence features are correlated to the epigenetic mark usage and the usage of enhancers of individual genes can have great variation, in this study, we focused on the genes and their promoter regions. With a more complete dataset on enhancers and their associated genes, we would expect a similar analysis will help understand the usage of enhancers of individual genes under different conditions. In fact, during T-cell differentiation, genes harboring multiple promoters possess more enhancers and enhancer diversity may contribute to the selective expression of isoforms (Maqbool et al, 2020). Aside from enhancers, other factors affecting the enhancer activity and enhancer–promoter interactions also influence the final gene expression pattern. For instance, in normal T cell, *MYC* and the corresponding super-enhancer resided in two TADs, respectively, insulated by CTCF. However, the loss of CTCF and consequently TAD fusion occurred in T-ALL (acute lymphoblastic leukemia), resulting in the establishment of spatial interactions between *MYC* and super-enhancer and the up-regulation of *MYC* (Kloetgen et al, 2020). In such a process, both CTCF and super-enhancer play vital roles in *MYC* expression.

Intriguingly, we found that for a small number of tissue-specific genes (c1-TSGs), the CpG density distribution around TSS is almost the same as that for housekeeping genes. Their difference appears to exist in TF binding: c1-TSGs are significantly depleted of TF-binding sites compared with c1-HKGs, which may prevent the former from being broadly expressed. We also noticed that unlike (general) tissue-specific genes, which are typically characterized by extremely low CpG density around TSS, genes encoding tissue-specific TFs generally possess high CpG densities (e.g., *FOXA1*). Such a sequence property may render a relative open environment for them to be easily accessed, as can be seen from MNase data, and to be expressed upstream of their target tissue-specific genes. A possible regulatory cascade therefore appears in which following the establishment of cell identity, housekeeping genes and genes encoding housekeeping TFs are activated early, after which genes encoding tissue-specific TFs become activated, and finally, the tissue-specific TFs access the promoter regions of tissue-specific genes to activate them (Fig S1C). Interestingly, the CpG density of

regions recruiting tissue-specific TFs is indeed significantly lower than that of the housekeeping TFs (Fig S17).

The tendency of gene expression changes during carcinogenesis also shows sequence biases as various types of cancers exhibit similar changes, including the preferred expression for high-CpG genes. Besides, the functions of dysregulated genes within one specific gene cluster are also similar for different types of cancers. Together with different modes of DNA methylation change for the three cluster genes, the similar epigenetic and expression changes among different types of cancer indicate that the mechanisms of cancer development are at least partially dictated by the sequence properties of genes, therefore pointing to the importance of including sequence properties in deciphering the cancer epigenomes.

# Materials and Methods

### RNN for gene classification

At first, all housekeeping genes (gene number is 1679) and tissue-specific genes (gene number is 1321) are labeled with 0 and 1, respectively, and then 800 housekeeping genes and 800 tissue-specific genes were chosen as the training set, and the remaining are regarded as the testing set. The RNN used in this study is based on bidirectional gate recurrent unit, with 1 layer and 16 hidden states, followed by a fully connected network with two hidden layers (the number of nodes is 16 and 2, respectively). Of note, in contrast with unlabeled genes (over 20,000 genes), the quantity of labeled genes is insufficient. Therefore, we used a data augmentation method called Unsupervised Data Augmentation (UDA) (Xie et al, 2019 *Preprint*) to make full use of the unlabeled genes. UDA encourages the model predictions to be consistent between an unlabeled example and an augmented unlabeled example. Thus, the loss function contained two parts: supervised and unsupervised term. The supervised term measured the binary cross entropy between the target $y$ and output $\hat{y}$, whereas the unsupervised term measured the clustering consistency between unlabeled original data and augmented data. The optimizer we used was Adam.

We then applied the well-trained network (which could distinguish the housekeeping gene from tissue-specific genes based on CpG density distribution) to the entire human gene set, and "CG likelihood" for each gene was obtained.

### The definition of tissue specificity

We downloaded the normalized gene expression data of 38 human tissues from https://zenodo.org/record/838734, and the tissue specificity of gene $i$ in tissue $t$, $s_i^t$, is calculated as (Tian et al, 2020)

$$s_i^t = \frac{\varepsilon_i^t - \mu_i^{all}}{\mu_i^{all}}$$

where $\varepsilon_i^t$ and $\mu_i^{all}$ are the mean expression levels of gene $i$ in tissue $t$ and all tissues, respectively.

### The calculation of the insulation score

For each 40-kb bin (the resolution corresponds to the resolution of Hi-C matrices in this study), its insulation score (Bintu et al, 2018) *IS* is calculated as

$$IS = \ln\left(1 + \frac{a1}{b} + \frac{a2}{b}\right)$$

where *a*1 and *a*2 are the average contact probabilities of its two flanking regions *A*1 and *A*2, respectively, and *b* is the average contact probability of the cross region *B* (the element of *B* represents the spatial interaction between *A*1 and *A*2). The window size we used here is 480 kb.

### The calculation of the Spearman correlation coefficient between the expression level and CpG/TpG density

For each gene cluster, the size of the corresponding CpG density matrix is $n \times 400$, where *n* represents the gene number in this cluster, and 400 is the window number. Accordingly, the size of expression vector in one specific cell is $n \times 1$. The Spearman correlation coefficient is then calculated between the expression vector and each column of CpG density matrix, yielding a $1 \times 400$ correlation coefficient vector. The correlations between TpG density and histone modifications, as well as between the expression level and histone modifications are calculated in the same way. For instance, as for Fig 3C, the ChIP-seq signal and DNA methylation level of each 40-bp window were calculated for each gene, resulting in a $n \times 400$ matrix for one epigenetic mark where n and 400 represent the gene number of one cluster and window numbers, respectively. The expression level matrix is $n \times 1$, and the correlation level between expression level matrix and each column of epigenetic mark matrix is finally calculated, yielding a $1 \times 400$ correlation vector for one cluster.

### The calculation of the compartment index

Based on the Hi-C contact matrix, the whole genome could be divided into two compartments, A and B (Lieberman-Aiden et al, 2009), with the former being more open and active, whereas the latter mainly corresponding to heterochromatin. The compartment index (*CI*) of bin is then calculated as

$$CI_i = \ln\left(\frac{C_{i-A}}{C_{i-B}}\right)$$

where $C_{i-A}$ and $C_{i-B}$ are the average normalized contact probabilities between bin *i* and compartment A bins and between bin *i* and compartment B bins, respectively. A higher value of *CI* thus indicates a more open environment.

### The identification of DE genes

During carcinogenesis, genes with $\log_2$ (expression fold change) > 1 and *P*-value < 0.05 are defined as up-expressed genes, and genes with $\log_2$(expression fold change) < −1 and *P*-value < 0.05 are regarded as down-expressed genes. Such results are obtained using DESeq2 (Love et al, 2014).

### Gene function analysis

The clusterProfiler package (Yu et al, 2012) was used in this study for gene function analysis. The background for GO analysis is all genes in orgDB by default.

## Data Availability

The data we analyzed in this work are publicly available and summarized in Table S1 (data sources).

## Supplementary Information

## Acknowledgments

We would like to thank Dr. Zhicheng Cai for insightful discussions. This research was supported by the National Natural Science Foundation of China (Nos. 92053202 and 22050003).

### Author Contributions

H Tian: conceptualization, formal analysis, visualization, methodology, and writing—original draft, review, and editing.
Y He: conceptualization, formal analysis, visualization, methodology, and writing—original draft.
Y Xue: conceptualization, formal analysis, visualization, methodology, and writing—original draft.
YQ Gao: conceptualization, supervision, funding acquisition, methodology, and writing—review and editing.

### Conflict of Interest Statement

The authors declare that they have no conflict of interest.

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
