## [Reviewer comments · Life Science Alliance]

Life Science Alliance

Expression regulation of genes is linked to their CpG density distributions around TSS

Hao Tian, Yueying He, Yue Xue, and Yi Gao
DOI: <https://doi.org/10.26508/lsa.202101302>

Corresponding author(s): Yi Gao, Peking University

Review Timeline:

Submission Date:	2021-11-17
Editorial Decision:	2022-01-11
Revision Received:	2022-03-29
Editorial Decision:	2022-05-05
Revision Received:	2022-05-07
Accepted:	2022-05-09

Scientific Editor: Novella Guidi

Transaction Report:

January 11, 2022

Re: Life Science Alliance manuscript #LSA-2021-01302

Prof. Yi Qin Gao
Peking University
Beijing National Laboratory for Molecular Sciences, College of Chemistry and Molecular Engineering
No. 5 Yiheyuan Road
Beijing 100871
China

Dear Dr. Gao,

Thank you for submitting your manuscript entitled "Understanding gene regulatory mechanisms based on gene classification" to Life Science Alliance. The manuscript was assessed by expert reviewers, whose comments are appended to this letter. We, thus, encourage you to submit a revised version of the manuscript back to LSA that responds to all of the reviewers' points.

Thank you for this interesting contribution to Life Science Alliance. We are looking forward to receiving your revised manuscript.

Sincerely,

B. MANUSCRIPT ORGANIZATION AND FORMATTING:

Reviewer #1 (Comments to the Authors (Required)):

In this manuscript Tian et al. perform gene classification based on CpG density distribution around the TSS and thoroughly characterise the relationship between promoter sequence properties and gene regulation patterns. Specifically, authors designed their own Recurrent Neural Network to classify genes based on CpG distribution patterns at their promoters. The authors investigated how CpG distribution around the TSS affects the regulatory landscape of the gene through a correlation analysis of the sequence properties with a large number of chromatin features, chromatin-associated proteins binding patterns and gene expression. The authors also made some interesting observations concerning distinct transcriptional and epigenetic changes of classified genes in cancer cells.

While gene classification based on entire CpG density/content across the gene promoter has been performed before (e.g. Weber et al. 2007; Mikkelsen et al. 2007), Tian et al. utilised the Recurrent Neural Network to classify gene promoters based on CpG distribution patterns rather than average values. While this new approach does not seem to offer much more in promoter sequence analysis and gene classification compared to previous work, this paper would be of interest to people interested in the role for sequence properties in such aspects of gene regulation as active and repressive chromatin marks deposition, 3D chromatin structure properties and gene expression.

Major comments:

1. How does the new classification of genes into clusters 1-3 compare to high, intermediate and low CpG promoter (HCP, ICP and LCP) (Weber et al. 2007; Mikkelsen et al. 2007). In addition to FigS1E-F it would be interesting to see the overlap between genes in each cluster and HCP, ICP and LCP genes previously identified.
2. Why has CpG density distribution of housekeeping vs tissue-specific gene promoters been used to train the network and obtain the 'CG likelihood'? Could it be done using CpG density distribution of CpG island vs nonCpG island promoters?
3. The Recurrent Neural Network used by authors for gene classification should be validated by comparing clusters 1-3 to computationally predicted CpG island datasets or BioCAP signal (human BioCAP data available from <https://www.ncbi.nlm.nih.gov/geo/query/acc.cgi?acc=GSE43512>)
It would be useful to include association of the promoter/entire gene with the CpG islands as well as compare lengths of the CpG clusters with CpG islands.
4. line 69: Was the gene length taken into account given that CpG density calculations have been performed across such a broad window (16kb)? Is the gene length distribution different across clusters?
5. line 109: The line plot does not convincingly show that the chromatin-associated proteins binding patterns become more dispersed. Perhaps a quantitative plot showing the distribution of the distances from the TSS to the binding site could be added?
6. Figures 2, 3 (C, D), S2, S3, S4, S5, S7 all contain similar correlation plots. Could authors please clarify the reason behind using spearman correlation plots? While these plots are useful, they hide a lot of information and the data would be clarified if there were additional figures included in supplementary files on the processed data used to generate these correlations, such as boxplots of normalised RNA-seq counts at genes within each cluster / subcluster of increasing CpG density (Fig 2). For the Fig 3 please add heatmaps of DNA methylation and histone marks such as H3K27ac where genes are sorted by the level of expression (similar to Fig 3A-B).
7. Additionally, the authors claim that different gene clusters exhibit distinct correlation patterns although the plots do not show it convincingly. Is it possible to perform a statistical test to support the claim?
8. Manuscript should include units for ChIP data (e.g. in Figure 3, it is unclear what the signal values refer to / what is being correlated, and it is not described in the methods)
9. Fig 4 - the authors indicate that cluster 1 genes tend to be upregulated while cluster 3 genes tend to be downregulated in

cancer. Are there significant differences between cluster's expression fold changes? Could authors please generate volcano plots (y-axis $\log(p\text{-value})$, x-axis fold change) for each gene cluster? Have the p-values been BH-adjusted to control for false discovery rate?

10. "Cluster 3 genes display a broad distribution of expression fold change, suggesting that some genes are dramatically activated/repressed in tumourigenesis". Could this rather be explained by the fact fold changes are often inflated for the lowly expressed genes in cluster 3? Unless authors performed the shrinkage of logFC estimates?

11. The DNA methylation changes result depicted on Figure 4D is interesting - please add a heatmap and SE to the line plot as line plots alone could be misleading with a small proportion of genes in the cluster driving the trend. How does this result agree with the literature?

Minor comments:

1. Figure 1C, 3C - please add SE to the line plot.
2. Figure 1D - please perform a statistical test
3. Figure 1H - I am sure how it the Figure 1H shows that CAPs tend to bind c1-HKG promoters
4. Line 172 - a typo: vertebrates instead of vertebrate
5. Figure 2: there is a typo on y-axis: should be level instead of evel
6. Line 266: to be uncoupled instead of uncouple
7. Line 273: these genes are intrinsically prone to silencing and their activation is - instead of these genes are intrinsically prone to silence and their activation are
8. What background was used for GO analysis?
9. "To gain information on the regulatory mechanism of cluster 3 genes, we calculated the correlation between compartment index (gene with higher value indicates it locates in a more compartment A environment, see Methods)" - manuscript should indicate that compartment A indicates euchromatin and compartment B indicates heterochromatin rather than just referring to methods, for clarity

Reviewer #2 (Comments to the Authors (Required)):

This manuscript, entitled "Understanding gene regulatory mechanisms based on gene classification" by Tian et al, describes the characterization of the genes classified by their novel scheme. The authors classified the human genes based on the CpG density around the TSS using a semi-supervised neural network method. Those genes were roughly classified into three groups. The authors found that the genes belonging to different clusters showed distinct properties in their expression levels and histone modification patterns, thus may indicate they receive distinct modes of gene expression regulations. They also investigated their possible roles in carcinogenesis and some features characteristic to each cluster in the context of carcinogenesis. Overall, I totally agree with the authors idea that the current CpG island-based method is already old-fashioned and development of a better method is desired. Now that large a volume of datasets is available for various omics layers, it is expected that further integration of the scattered pieces of information on the CpG island and histone modification patterns with consequential transcriptome patterns would shed a new light on further biologically relevant understandings of global gene expression regulations. However, I'm not convinced that this paper should make a substantial contribution towards that goal. Particularly, some parts of gene expression regulations should have inherently unique features, while other parts of them should be shared with other members of the genes belonging to the same regulatory network. The former unique features are left almost totally uncharacterized, which should limit the biological interpretation of the obtained results.

Major points:

1. I do not understand the biological relevance of this particular paper. Neural network can separate the features of any kinds to any sub-groups. Assuming each of the separate groups may correspond to various biological features, then, what does the classification indicate from a biological viewpoint? Without extensive analyses for the factors contributed to the classifications, the present analysis is not more than a description to the arbitrary separated groups.
2. Alternatively, even without obtaining a particular biological indication, is not this classification useful for some biological purposes, for example, for predicting the cell fate or cancer diagnosis?
3. If the authors claim the methodological novelty in the presented method or the usefulness of the obtained classes themselves, they have to show their advantages over the classical CpG ration and GC contents-based method or CpG islands.

Further precise evaluation should be needed how the correlation may differ between the new approach and the classical approach, particularly regarding their association with the epigenomic features.

4. I'm also concerned the present analyses completely overlook the enhancers, which I believe should play no less important roles in gene expression regulations.

5. The strategy to make use of the recurrent neural network (RNN) is not clear. Judging from the descriptions in the Materials and Methods, it is designed simply to separate "housekeeping" and "tissue-specific" genes based on the CpG features. I wonder if the following analyses just represent the associations between the tissue-specificity and the epigenome features, not necessarily associated with the utilized CpG features. Moreover, the definition of the housekeeping and tissue-specific genes is an issue of discussion. Please note that the widely expressed genes (or even ubiquitously expressed genes) may not always be the genes of housekeeping genes in their functional aspects, especially when their expression levels are not quantitatively analyzed.

Minor points:

6. Overall, the paper should be difficult to read for general biologists. Without exemplifying the regulation mechanisms of particular genes in the text and figures, it is difficult to follow the intention of the authors.

7. The employed dataset seems arbitrary and limited. At least, it is not presented in a well-ordered manner. Particular focus seems to be put on the HepG2 dataset, but this cell type should represent only a limited features of the wide variety of cell types.

8. Assuming the CpG features are evolutionarily conserved, I wonder how the genes which receive distinct regulation between humans and mice would look like in this study.

9. For the cancer story, it remains elusive how the gene regulation of each class should be associated with global hyper- or hypo-methylation of the cancers, which is highly variable depending on cancer types and stages.

10. Abstract is superficial and does not summarize the contents of the paper.

11. Abstract: line 2: 3D structure is not the central issue of this paper, thus, should be removed.

12. The last paragraph of the Introduction section should be redundant to the descriptions in the Results section.

13. Discussion is not so much helpful to deepen the interpretation of the obtained results in this particular study. It is rather summarizing the contents of the previous section.

14. Title is ambiguous and does not represent the subject of the paper, although the subject itself still remains blur to me.

Reviewer #1 (Comments to the Authors (Required)):

In this manuscript Tian et al. perform gene classification based on CpG density distribution around the TSS and thoroughly characterise the relationship between promoter sequence properties and gene regulation patterns. Specifically, authors designed their own Recurrent Neural Network to classify genes based on CpG distribution patterns at their promoters. The authors investigated how CpG distribution around the TSS affects the regulatory landscape of the gene through a correlation analysis of the sequence properties with a large number of chromatin features, chromatin-associated proteins binding patterns and gene expression. The authors also made some interesting observations concerning distinct transcriptional and epigenetic changes of classified genes in cancer cells.

While gene classification based on entire CpG density/content across the gene promoter has been performed before (e.g. Weber et al. 2007; Mikkelsen et al. 2007), Tian et al. utilised the Recurrent Neural Network to classify gene promoters based on CpG distribution patterns rather than average values. While this new approach does not seem to offer much more in promoter sequence analysis and gene classification compared to previous work, this paper would be of interest to people interested in the role for sequence properties in such aspects of gene regulation as active and repressive chromatin marks deposition, 3D chromatin structure properties and gene expression.

Major comments:

1. How does the new classification of genes into clusters 1-3 compare to high, intermediate and low CpG promoter (HCP, ICP and LCP) (Weber et al. 2007; Mikkelsen et al. 2007). In addition to FigS1E-F it would be interesting to see the overlap between genes in each cluster and HCP, ICP and LCP genes previously identified.

We thank the reviewer for this suggestion. We calculated the overlap between genes in each cluster identified in this study and HCP, ICP and LCP genes previously identified. The results are displayed below and clearly revealed that most HCP and LCP genes belong to cluster 1 and 3, respectively, whereas ICP is distributed among all three classes. The results have been added in the supplementary materials.

2. Why has CpG density distribution of housekeeping vs tissue-specific gene promoters been used to train the network and obtain the CG likelihood? Could it be done using CpG density distribution of CpG island vs nonCpG island promoters?

In this work, we mainly focused on how gene regulatory mechanisms are related to its CpG density distribution around TSS. And many studies have already revealed that housekeeping genes and tissue-specific genes harbor distinctly different CpG density

distributions (Serge et al., PNAS, 2006). As can be seen in our work, housekeeping genes tend to possess a sharp and high CpG density peak around TSS (cluster 1) while tissue-specific genes are generally characterized by extremely low CpG density around TSS (cluster 3). Hence, we used the CpG density of housekeeping and tissue-specific genes to train our network and the output of the network indicated the CpG density distribution of the individual genes. One important discovery here is cluster 2, the genes of which harbor broad and high CpG density peaks, and the expression level of these genes are most correlated with H3K27me3 signal.

The CpG density distribution of CpG island promoters is complicated due to the variation of CGI length, CGI position and CG content. It is not proper to regard CpG island promoters as one group, the labels within which is the same.

3. The Recurrent Neural Network used by authors for gene classification should be validated by comparing clusters 1-3 to computationally predicted CpG island datasets or BioCAP signal (human BioCAP data available from <https://www.ncbi.nlm.nih.gov/geo/query/acc.cgi?acc=GSE43512>) It would be useful to include association of the promoter/entire gene with the CpG islands as well as compare lengths of the CpG clusters with CpG islands.

We thank the reviewer for these suggestions. We firstly compared the clusters 1-3 with CGI datasets, and as expected, almost all cluster 1 genes are associated with CGI whereas only a minority of cluster 3 genes are associated with CGI (Figure 3-1). Such an association was also observed when we compared genes of three clusters with non-methylated islands (NMI) identified by Bio-CAP (Figure 3-2). Besides, CGI/liver NMI tend to be associated with housekeeping genes (Figure 3-3 and 3-4). The results have been added in the manuscript.

Figure 3-1 The proportion of genes whose promoter regions overlapped with CGI in

each cluster.

Figure 3-2 The proportion of genes whose promoter regions overlapped with NMI in each cluster.

Figure 3-3 The overlap between CGI/non-CGI and housekeeping genes/tissue-specific genes.

Figure 3-4 The overlap between liver NMI/MI and housekeeping genes/tissue-specific genes.

4. line 69: Was the gene length taken into account given that CpG density calculations have been performed across such a broad window (16kb)? Is the gene length distribution different across clusters?

In the classification of genes, we did not consider the gene length. To generate a more general model, we chose a broad window (16 kb). The results we obtained are consistent with those using an 8 kb window.

The gene length varies from $\sim 10^3$ bp to $\sim 10^5$ bp and different clusters did not show gene length distribution difference (please see figure below).

5. line 109: The line plot does not convincingly show that the chromatin-associated proteins binding patterns become more dispersed. Perhaps a quantitative plot showing the distribution of the distances from the TSS to the binding site could be added?

We thank the reviewer for this suggestion. The line in figure 1E indeed represents the probability density distribution of the distance between TSS and the binding site. For clarity, we performed statistic test between cluster 1 and clusters 2/3 (cluster 1 VS cluster 2, p-value = 2.18e-09; cluster 1 VS cluster 3, p-value = 0.04) and the corresponding p-values have been added in the manuscript.

6. Figures 2, 3 (C, D), S2, S3, S4, S5, S7 all contain similar correlation plots. Could authors please clarify the reason behind using spearman correlation plots? While these plots are useful, they hide a lot of information and the data would be clarified if there were additional figures included in supplementary files on the processed data used to generate these correlations, such as boxplots of normalised RNA-seq counts at genes within each cluster / subcluster of increasing CpG density (Fig 2). For the Fig 3 please add heatmaps of DNA methylation and histone marks such as H3K27ac where genes are sorted by the level of expression (similar to Fig 3A-B).

We thank the reviewer for these helpful suggestions.

Spearman correlation has been widely applied in analyzing bio-omics data (Yang et al., BMC Genomics, 2014; Josie et al., Nucleic Acids Research, 2022; Takuya et al., Oncology Letters, 2022;) since within which, some values (such as expression levels or CpG densities of several genes) could be overly large and bias the Pearson calculation.

The rank-based Spearman correlation helps alleviate such a problem and was thus used here.

Based on reviewer's suggestion, the expression levels of the three clusters in 2-cell, 8-cell, ICM and hESC have been added in supplementary materials. And the heatmaps of DNA methylation and H3K27ac sorted by expression level have also been added in corresponding supplementary figures.

7. Additionally, the authors claim that different gene clusters exhibit distinct correlation patterns although the plots do not show it convincingly. Is it possible to perform a statistical test to support the claim?

We thank the reviewer for this helpful suggestion. We performed statistic test on correlation patterns and the corresponding p-values are generally small. For instance, as for the correlation patterns between gene expression level and CpG density, the p-values are 5.00e-31 (cluster1 VS cluster2 for 2-cell), 1.04e-133 (cluster1 VS cluster3 for 2-cell) and 9.92e-177 (cluster2 VS cluster3 for 2-cell), and all p-values $< 10^{-7}$ in Figure 3C, which indicate that different gene clusters do exhibit distinct correlation patterns.

The corresponding p-values have been added in the manuscript.

8. Manuscript should include units for ChIP data (e.g. in Figure 3, it is unclear what the signal values refer to / what is being correlated, and it is not described in the methods)

We thank the reviewer for this kind suggestion. The ChIP data was downloaded in fold-change format (ChIP-seq counts relative to control) from Roadmap. Such information has been added to the figure legends. The correlation in Figure 3C was calculated as follows: the ChIP-seq signal and DNA methylation level of each 40 bp window were calculated for each gene, resulting in a $n \times 400$ matrix for one epigenetic mark where n and 400 represent the gene number of one cluster and window numbers, respectively. The expression level matrix is $n \times 1$, and the correlation level between expression level matrix and each column of epigenetic mark matrix is finally calculated, yielding a 1×400 correlation vector for one cluster. For clarity, the calculation method has been added in the Methods section.

9. Fig 4 - the authors indicate that cluster 1 genes tend to be upregulated while cluster 3 genes tend to be downregulated in cancer. Are there significant differences between clusters expression fold changes? Could authors please generate volcano plots (y-axis $\log(p\text{-value})$, x-axis fold change) for each gene cluster? Have the p-values been BH-adjusted to control for false discovery rate?

We thank the reviewer for this thoughtful suggestion. We performed statistical test between expression fold change of cluster 1 and cluster 3 genes and the corresponding p-values have been added in the manuscript. All p-values are less than 0.001 by Welch's unequal variance t-test for 6 types of cancers. Besides, we have added volcano plots for

each gene cluster in supplementary materials. All p-values have been BH-adjusted.

10. "Cluster 3 genes display a broad distribution of expression fold change, suggesting that some genes are dramatically activated/repressed in tumourigenesis". Could this rather be explained by the fact fold changes are often inflated for the lowly expressed genes in cluster 3? Unless authors performed the shrinkage of logFC estimates?

We thank the reviewer for this suggestion. We performed the shrinkage of logFC estimates and the observations remain the same (please see figures below), except for STAD within which the fold changes of majority of genes are shrunk to zero (Figure 10-1, 10-2). The shrinkage has little impact on the genes which are significantly up-/down-regulated (Figure 10-1). And the proportion of significantly changed genes is relatively larger for cluster 3 compared to cluster 1 and 2 (Figure 10-3), accordant with our previous conclusion. The above results have been added in the manuscript.

Figure 10-1 The volcano plots obtained from DESeq2 analysis without shrinkage (the upper figure) and with shrinkage (the bottom figure).

Figure 10-2 The boxplots for expression fold-change after shrinkage.

Figure 10-3 The proportion of DE genes (in carcinogenesis) identified based on shrunken fold changes in three clusters. Upper: the proportion of up-expressed genes, down: the proportion of down-expressed genes.

11. The DNA methylation changes result depicted on Figure 4D is interesting - please add a heatmap and SE to the line plot as line plots alone could be misleading with a small proportion of genes in the cluster driving the trend. How does this result agree with the literature?

We thank the reviewer for this suggestion. We have added SE on Figure 4D, as well as corresponding heatmaps in supplementary materials (shown below). The heatmaps show consistent conclusions compared to line plots.

In this work, we found that the TSS of cluster 2 genes tend to be hypermethylated in carcinogenesis. At the same time, cluster 2 genes are also significantly enriched with H3K27me3 in corresponding normal tissues, consistent with the findings that most promoters that exhibit cancer-associated hypermethylation are linked to genes silenced by polycomb and H3K27me3 in corresponding normal tissues (Ehrlich et al., Epigenetics, 2019; Schlesinger et al., Nature Genetics, 2006). For hypomethylation, our previous study (Xue et al., Molecular Oncology, 2022) found that loci with low CpG density located in repressive and inaccessible environment (compartment B) and are more likely to be hypomethylated in cancer cells. Consistently, we found here that cluster 3 genes have the lowest CpG densities, expression and accessibility, therefore they are more likely to be hypomethylated compared to genes in the other two clusters. We have added the above discussion and related literatures to the manuscript.

Minor comments:

1. Figure 1C, 3C - please add SE to the line plot.

We thank the reviewer for this suggestion and the SE has been added to Figures 1C and 1G. As we mentioned in comment 8, Figure 3C represents the correlation level between expression and epigenetic marks for genes and SE was not calculated.

2. Figure 1D - please perform a statistical test

The statistical test has been performed and the corresponding p-values have been added in the figure legend (cluster 1 vs cluster 2: p-value = 3.12e-69; cluster 1 vs cluster 3: p-value = 6.37e-276; cluster 2 vs cluster 3: p-value = 4.28e-106).

3. Figure 1H - I am sure how it the Figure 1H shows that CAPs tend to bind c1-HKG promoters

There are 208 data points (corresponding to 208 chromatin associated proteins) in Figure 1H and each data point represents the proportion of c1-HKGs (x-axis) and h-c1-TSGs (y-axis) binding to one certain TF. Such a proportion tends to be higher for c1-HKGs (p -value = $4.12e-11$), thus indicating the higher capacity of TF binding for c1-HKGs.

4. Line 172 - a typo: *vertebrates* instead of *vertebrate*

Correction has been made.

5. Figure 2: there is a typo on y-axis: *should be level* instead of *evel*

Typo has been fixed.

6. Line 266: *to be uncoupled* instead of *uncouple*

Correction has been made.

7. Line 273: *these genes are intrinsically prone to silencing and their activation is - instead of these genes are intrinsically prone to silence and their activation are*

The word has been adjusted.

8. *What background was used for GO analysis?*

The background for GO analysis is all genes in orgDB by default. We have added the explanation in the Method section.

9. *"To gain information on the regulatory mechanism of cluster 3 genes, we calculated the correlation between compartment index (gene with higher value indicates it locates in a more compartment A environment, see Methods)" - manuscript should indicate that compartment A indicates euchromatin and compartment B indicates heterochromatin rather than just referring to methods, for clarity*

We thank the reviewer for this helpful suggestion and for clarity, "compartments A and B are mainly corresponding to euchromatin and heterochromatin, respectively" has been added to the corresponding position of the manuscript.

Reviewer #2 (Comments to the Authors (Required)):

This manuscript, entitled "Understanding gene regulatory mechanisms based on gene classification" by Tian et al, describes the characterization of the genes classified by their novel scheme. The authors classified the human genes based on the CpG density around the TSS using a semi-supervised neural network method. Those genes were roughly classified into three groups. The authors found that the genes belonging to different clusters showed distinct properties in their expression levels and histone modification patterns, thus may indicate they receive distinct modes of gene expression regulations. They also investigated their possible roles in carcinogenesis and some features characteristic to each cluster in the context of carcinogenesis. Overall, I totally agree with the authors idea that the current CpG island-based method is already old-fashioned and development of a better method is desired. Now that large a volume of datasets is available for various omics layers, it is expected that further integration of the scattered pieces of information on the CpG island and histone modification patterns with consequential transcriptome patterns would shed a new light on further biologically relevant understandings of global gene expression regulations. However, Im not convinced that this paper should make a substantial contribution towards that goal. Particularly, some parts of gene expression regulations should have inherently unique features, while other parts of them should be shared with other members of the genes belonging to the same regulatory network. The former unique features are left almost totally uncharacterized, which should limit the biological interpretation of the obtained results.

Major points:

1. I do not understand the biological relevance of this particular paper. Neural network can separate the features of any kinds to any sub-groups. Assuming each of the separate groups may correspond to various biological features, then, what does the classification indicate from a biological viewpoint? Without extensive analyses for the factors contributed to the classifications, the present analysis is not more than a description to the arbitrary separated groups.

We thank the reviewer for this comment. The main purpose of this study is to investigate whether relatively simple sequential features of a gene are associated with their expression patterns in different cells and whether these sequence features are correlated to the epigenetic mark usage. In fact, the interplay between epigenetic marks (such as histone modification and DNA methylation), 3D chromatin structure and gene regulation is still far from fully understood. For instance, large amounts of studies have revealed that H3K4me3 and H3K9me3 are associated with gene activation and repression (Ren et al., Journal of Hematology & Oncology, 2021; Ninova et al., Development, 2019), respectively, however, at the same time, there exists many studies revealing that histone modification seems to be uncorrelated with gene expression (Murray et al., bioRxiv, 2019; Borsari et al., bioRxiv, 2020). On the other hand, the distribution of CpG density has been found to be

associated with high order chromatin structure as well as the “deposition” of histone marks, one example is CG-rich sequence could recruit PRC2, catalyzing the tri-methylation of H3K27. Thus, it appears that sequence-dependence of certain epigenetic mark usage widely exist, e.g., some CG-rich genes could inherently recruit PRC2 for repression, resulting in the high correlation between gene regulation and histone modification, while genes with low CG density tend to lack histone marks and their activation relied more on specific TF binding, and thereby exhibited low correlation between histone mark and gene expression. In other words, the controversy introduced above may at least in part derive from the diversity of CpG density distribution. In addition, there is an obvious CpG-density dependence on DNA methylation and related expression regulation (please also see reply to point 2). Therefore, to gain insights on the underlying sequence-dependence on the epigenetic modification, we tried to perform gene classification based on CpG density distribution around TSS and subsequently investigate the differences of gene regulatory mechanisms behind different gene clusters, which help us at least partially understand why the relation between epigenetic, gene regulation and chromatin structure appears to be complex.

2. Alternatively, even without obtaining a particular biological indication, is not this classification useful for some biological purposes, for example, for predicting the cell fate or cancer diagnosis?

As we mentioned in point 1, the appearing controversial relation between epigenetics, chromatin structure and gene regulation may derive from the diverse CpG density distribution in different genes and we thus performed gene classification based on CpG density distribution. We then found that different gene clusters indeed exhibited distinct regulatory mechanisms, for instance, the regulation of cluster 3 genes relied less on epigenetics and more on chromatin reorganization. The analyses therefore showed that it is important to consider CpG density distribution around TSS to understand gene regulatory mechanisms.

Another point we emphasized in this manuscript is the similar sequence-dependent variation among different tumors. For instance, cluster 1 and 3 genes tend to be up- and down-regulated during tumorigenesis, respectively, the TSS of cluster 2 genes undergo the most significant hypermethylation among three clusters, and the up- and downstream of TSS of cluster 3 genes tend to be more hypomethylated than other genes. Such a trend was observed in different cancer types, indicating that the mechanism of cancer development is at least partially dictated by the sequence property, thus pointing to the importance of including sequence properties in deciphering and predicting the cancer epigenomes.

3. If the authors claim the methodological novelty in the presented method or the usefulness of the obtained classes themselves, they have to show their advantages over the classical CpG ration and GC contents-based method or CpG islands. Further precise evaluation should be needed how the correlation may differ between the new approach and the classical approach, particularly regarding

their association with the epigenomic features.

We thank the reviewer for this helpful suggestion. As can be seen from Figures S1H and S1I, the variation of CpG density distribution and tissue specificity within one group is smaller in clusters identified in this work, compared to HCP, LCP and ICP, indicating the “homogeneity” of our cluster. Besides, for HCP, ICP and LCP, we also calculated the correlation between expression level and epigenetic marks. Interestingly, the correlation between expression and H3K27me3 is significantly stronger in cluster 2, compared to ICP (please see figure below). This observation is consistent with cluster 2 genes being characterized by broad and high CpG density peak which are prone to the trimethylation of H3K27 for repression. Interestingly, this class of genes do show a stronger correlation between expression and H3K27me3 than the other classes. The above results have been added in the manuscript.

4. I'm also concerned the present analyses completely overlook the enhancers, which I believe should play no less important roles in gene expression regulations.

We thank the reviewer for this comment. We totally agree with the reviewer on the importance of enhancers. Enhancers have been suggested to play vital roles in gene regulation given that they provide accommodations for transcription factors and could form spatial contact with promoters to activate specific genes. In general, H3K27ac/H3K4me1/ATAC/DNase peaks were regarded as candidate enhancers (Elizabeth et al., Nature Genetics, 2021; Giuliano et al., Nature Communications, 2021; Rurika et al., Genome Biology, 2017). Then the candidate enhancer nearest to genes was thought to regulate that gene. However, with the development of chromosome conformation capture technology, many studies have revealed that enhancer could skip its nearest gene and regulate genes far away from that enhancer. Akin to this, our work revealed that the activation of cluster 3 genes (mainly residing in heterochromatin) relied more on movement toward compartment A, indicating that the enhancers regulating cluster 3 genes may be mainly in compartment A (akin to the phase separation mechanisms (Hnisz et al., Cell, 2017)), suggesting the longer genome distance between cluster 3 genes and its

corresponding enhancer, compared to cluster 1 and 2 genes. As mentioned earlier, since the main purpose of this study is to investigate whether relatively simple sequential features of a gene are associated with their expression patterns in different cells and whether these sequence features are correlated to the epigenetic mark usage, and the usage of enhancers of individual genes can have great variation, in this study we focused on the genes and their promoter regions. With a more complete dataset on enhancers and their associated genes, we would expect a similar analysis will help understand the usage of enhancers of individual genes under different conditions.

The above discussion has been added in the manuscript.

5. The strategy to make us of the recurrent neural network (RNN) is not clear. Judging from the descriptions in the Materials and Methods, it is designed simply to separate "housekeeping" and "tissue-specific" genes based on the CpG features. I wonder if the following analyses just represent the associations between the tissue-specificity and the epigenome features, not necessarily associated with the utilized CpG features. Moreover, the definition of the housekeeping and tissue-specific genes is an issue of discussion. Please note that the widely expressed genes (or even ubiquitously expressed genes) may not always be the genes of housekeeping genes in their functional aspects, especially when their expression levels are not quantitatively analyzed.

Previous studies have already revealed that housekeeping genes and tissue-specific genes possess distinctly different CpG density distributions (Serge et al., PNAS, 2006), and as can be seen from Figure 1 (in the manuscript), housekeeping genes harbor a high and sharp CpG density peak around TSS while tissue-specific genes are characterized by extremely low CpG density. Such distinct features (that is, the two groups) provide opportunities for us to train a neural network, the similar output values of which indicate the similar CpG density distributions. Hence, based on the output values (named "CG likelihood"), we divided the human genes into three clusters, and genes within one cluster share similar CpG density distribution (Figure 1C in manuscript). The three gene clusters with distinctly different CpG density distributions further provide opportunities to investigate how regulatory mechanisms differ among the three clusters.

In this study, the housekeeping gene list was downloaded from <https://www.tau.ac.il/~elieis/HKG> and tissue-specific genes were quantitatively defined based on the GTEx expression data. Housekeeping and tissue-specific genes were further validated using CAGE data (<https://fantom.gsc.riken.jp/5/>). As can be seen below, housekeeping genes tend to be highly expressed in almost all tissues (genes with expression level higher than 10 is defined as highly expressed gene) while tissue-specific genes are highly and specifically expressed in only several tissues. In fact, during the network training, we did not consider their functional aspects and only utilized their distinctly different CpG density distributions.

Minor points:

6. Overall, the paper should be difficult to read for general biologists. Without exemplifying the regulation mechanisms of particular genes in the text and figures, it is difficult to follow the intention of the authors.

We thank the reviewer for this comment. As we mentioned in the manuscript, cluster 1 and 3 genes always tend to be up- and down-regulated during tumorigenesis of different cancer types, and the TSS of cluster 2 genes always tend to be hypermethylated. Following review's suggestion, we added several examples (related to gene regulatory mechanisms). For instance, *SLC16A1* and *BSG*, belonging to cluster 1 and playing vital roles in energy metabolism, are up-regulated in a variety of cancer types and could promote tumor growth and aggressiveness through Warburg effect. Besides, *GATA4*, *HOXD11* and *HOXD12*, belonging to cluster 2 and participating in development and morphogenesis, are hypermethylated in various cancer types.

7. The employed dataset seems arbitrary and limited. At least, it is not presented in a well-ordered manner. Particular focus seems to be put on the HepG2 dataset, but this cell type should represent only a limited features of the wide variety of cell types.

In this paper, when we calculated the correlation between expression level and CpG/TpG density, we used many samples in different stages, (for instance, four embryonic-related samples (2-cell, 8-cell, ICM and hESC), 10 somatic tissues, and 20 tumor and corresponding normal tissues). For the cancer analysis, we also used six samples with complete data. Different samples yielded similar results, indicating the reliability of our conclusions. The data we used in this work were also summarized in additional file 2.

Since the ChIP-seq experiments of 208 chromatin-associated proteins were performed in HepG2 cell (Partridge et al., Nature, 2020), therefore, we used HepG2 expression data to define h-c1-TSG, and the TF binding ability of which was further compared with c1-HKGs.

8. Assuming the CpG features are evolutionarily conserved, I wonder how the genes which receive distinct regulation between humans and mice would look like in this study.

We compared the CpG density distributions of housekeeping and tissue-specific genes between human and mice and observed similar properties between the two species (please see figures below). Besides, the association between gene regulatory mechanisms and CpG density distribution is conserved between human and mice (Kruijsbergen et al., Int J Biochem Cell Biol., 2015). For instance, CpG-rich regions were found to recruit PRC2 for H3K27me3 deposition in both human and mice cells (Mendenhall et al., PLoS Genetics, 2010; Lynch et al., EMBO J., 2012). Therefore, we speculated that genes receive distinct regulation between human and mice harbor distinct CpG density distributions.

9. For the cancer story, it remains elusive how the gene regulation of each class should be associated with global hyper- or hypo-methylation of the cancers, which is highly variable depending on cancer types and stages

This is a helpful suggestion. In this paper, we found that cluster 2 genes are significantly enriched in H3K27me3 in normal tissues and hypermethylated during tumorigenesis, and many studies have also revealed that regions decorated by H3K27me3 tend to be hypermethylated in carcinogenesis (Ehrlich, Epigenetics, 2019; Schlesinger et al., Nature Genetics, 2006). In fact, the hypermethylation loci is highly conserved among different types of cancer and stages (please see figure below, $p < 10^{-300}$ by Fisher exact exam between read2 and blca6). For hypomethylation, in our previous research (Xue et al., Molecular Oncology, 2022), we found that loci with low CpG density located in repressive and inaccessible environment (compartment B) and tend to be hypomethylated in cancer cells. In this work we found that cluster 3 genes have low CpG densities and accessibility, therefore the up-/down-stream of cluster 3 genes are more likely to be hypomethylated compared to other clusters.

10. *Abstract is superficial and does not summarize the contents of the paper.*

We thank the reviewer for this comment. The abstract has been modified to better summarize the contents of the paper.

11. *Abstract: line 2: 3D structure is not the central issue of this paper, thus, should be removed.*

The “3D genome organization” has been removed accordingly.

12. *The last paragraph of the Introduction section should be redundant to the descriptions in the Results section.*

We thank the reviewer for this kind comment and we have refined the last paragraph of introduction.

13. *Discussion is not so much helpful to deepen the interpretation of the obtained results in this particular study. It is rather summarizing the contents of the previous section.*

We thank the reviewer for this comment. Accordingly, we added in the discussion the contents related to enhancers and the comparison of correlation levels between clusters identified in this work and previous work (Weber et al., Nature Genetics, 2007).

14. *Title is ambiguous and does not represent the subject of the paper, although the subject itself still remains blur to me.*

The title has been modified as “Expression regulation of genes is linked to their CpG density distributions around TSS”.

May 5, 2022

RE: Life Science Alliance Manuscript #LSA-2021-01302R

Prof. Yi Qin Gao
Peking University
Beijing National Laboratory for Molecular Sciences, College of Chemistry and Molecular Engineering
No. 5 Yiheyuan Road
Beijing 100871
China

Dear Dr. Gao,

Thank you for submitting your revised manuscript entitled "Expression regulation of genes is linked to their CpG density distributions around TSS". We would be happy to publish your paper in Life Science Alliance pending final revisions necessary to meet our formatting guidelines.

- please address the remaining Rev 1 points
- please address the remaining Rev 2 concerns through changes in the the text and deeming down the analysis of the enhancers
- please upload your manuscript text as an editable doc file
- please upload your supplementary figure files as single files and add the supplementary figure legends to your main manuscript text, directly under the main figure legends
- please add the Twitter handle of your host institute/organization as well as your own or/and one of the authors in our system
- please add a callout for Figure S6 & Figure S15A-C in your main manuscript text

Figure Issues:

- please adjust the order of panels in Figure S1
- please expand legends and further explain panels in supplementary figures S2,3,4,5,6,8,11,12,13,16,17

A. FINAL FILES:

B. MANUSCRIPT ORGANIZATION AND FORMATTING:

Sincerely,

Reviewer #1 (Comments to the Authors (Required)):

While some of the figures (particularly the over-abundant Spearman correlation plots) remain rather difficult to interpret and extract biological sense from, the authors have performed substantial work and additional analyses to address reviewers' comments. In general, I am satisfied with the way my questions were addressed and recommend, in principle, Tian et al manuscript for publication after the following comments have been addressed.

Minor comments:

The abstract contains no conclusive information, it only states that different clusters exhibit different features (e.g "these different clusters exhibited distinctly different structural features, regulatory mechanisms, ...") without any specific conclusions. Could authors please describe the actual SPECIFIC findings in the abstract?

Fig S6 is not mentioned in the text. Please discuss the difference in gene expression between clusters and the difference in spearman correlation you observed and discuss the potential biology behind it. E.g why the cluster 1 genes with the highest gene expression display the lowest correlation with CpG density?

Fig S7E - a typo: expression instead of "expre"

Reviewer #2 (Comments to the Authors (Required)):

First of all, I appreciate the substantial efforts of the authors for the revision of this manuscript. I admit that the manuscript has been very much improved by the extensive analyses and the deepened discussion, However, I have to point out that the most important issue of mine that I have pointed out in my previous review has been only poorly addressed. As the authors described in the Response 4, it is true that "the usage of enhancers of individual genes can have great variation". However, because of this reason, further specific analyses, where the analyses of enhancers should be also included, should be of the greatest importance. This diversity may be even larger for the regulations of "key" genes of various biological contexts, rather than "housekeeping genes". In those genes, to realize the complex regulations depending on variable tissues and cellular statuses, it

is even possible that the "core" promoters may determine just basic levels of the gene expression levels in a "passive" manner and the enhancers may play rather pivotal roles. In fact, increasing evidence shows that the modules of enhancers, which are not always located in the distal regions of the gene bodies, have essential roles in various biological contexts, including diseases. For the challenge to include the enhancers into the consideration, substantial, even not complete, functional catalogues of enhancers have been available. Above is the issue for the enhancers, just for instance, but there should be many other factors determining eventual gene expression "patterns" in a given biological environment. Without considering those factors, depending on a specific gene in a particular biological context, at least for some representative cases, the relevance of this study should no more a re-visit of the classical categorization of the promoters.

Reviewer #1 (Comments to the Authors (Required)):

While some of the figures (particularly the over-abundant Spearman correlation plots) remain rather difficult to interpret and extract biological sense from, the authors have performed substantial work and additional analyses to address reviewers comments. In general, I am satisfied with the way my questions were addressed and recommend, in principle, Tian et al manuscript for publication after the following comments have been addressed.

Minor comments:

1. The abstract contains no conclusive information, it only states that different clusters exhibit different features (e.g "these different clusters exhibited distinctly different structural features, regulatory mechanisms, ...") without any specific conclusions. Could authors please describe the actual SPECIFIC findings in the abstract?

We thank the reviewer for this comment. The abstract has been modified accordingly, and especially the detailed and concrete findings have been added.

2. Fig S6 is not mentioned in the text. Please discuss the difference in gene expression between clusters and the difference in spearman correlation you observed and discuss the potential biology behind it. E.g why the cluster 1 genes with the highest gene expression display the lowest correlation with CpG density?

We thank the reviewer for pointing out it. As mentioned in this manuscript and reported by our previous studies (Liu et al., *Nucleic Acids Research*, 2018), the promoter regions of cluster 1 genes tend to locate in an open environment and cluster 1 genes mainly resided in compartment A, resulting in their highest expression level among the three gene clusters. Benefiting from the unique chromatin environment (possibly due to their unique sequence features), cluster 1 genes are more likely to stably express (almost all housekeeping genes are in cluster 1), thus contributing to the low correlation between expression level and CpG density.

The above discussion has been added to the manuscript.

3. Fig S7E - a typo: expression instead of "expre"

We have modified it.

Reviewer #2 (Comments to the Authors (Required)):

First of all, I appreciate the substantial efforts of the authors for the revision of this manuscript. I admit that the manuscript has been very much improved by the extensive analyses and the deepened discussion, However, I have to point out that the most important issue of mine that I have pointed out in my previous review has been only poorly addressed. As the authors described in the Response 4, it is true that "the usage of enhancers of individual genes can have great variation". However, because of this reason, further specific analyses, where the analyses of enhancers should be also included, should be of the greatest importance. This diversity may be even larger for the regulations of "key" genes of various biological contexts, rather than "housekeeping genes". In those genes, to realize the complex regulations depending on variable tissues and cellular statuses, it is even possible that the "core" promoters may determine just basic levels of the gene expression levels in a "passive" manner and the enhancers may play rather pivotal roles. In fact, increasing evidence shows that the modules of enhancers, which are not always located in the distal regions of the gene bodies, have essential roles in various biological contexts, including diseases. For the challenge to include the enhancers into the consideration, substantial, even not complete, functional catalogues of enhancers have been available. Above is the issue for the enhancers, just for instance, but there should be many other factors determining eventual gene expression "patterns" in a given biological environment. Without considering those factors, depending on a specific gene in a particular biological context, at least for some representative cases, the relevance of this study should no more a re-visit of the classical categorization of the promoters.

We thank the reviewer for these suggestions. As nicely pointed out by the reviewer, the diversity of enhancer usage may be large for the regulation of "key" genes, which depends on the cellular states, and aside from enhancer activity, other factors affecting the enhancer activity and enhancer-promoter interactions also influence the final gene expression pattern. For instance, during T cell differentiation, genes harboring multiple promoters contain more enhancers and enhancer diversity may contribute to selecting isoform expression (Maqbool et al., Cell Reports, 2020). Another example is the regulation of *MYC* (Andreas et al., Nature Genetics, 2020), which has been shown to be closely associated with tumorigenesis. In normal T cell, *MYC* and the corresponding super-enhancer resided in two TADs, respectively, insulated by CTCF. However, the loss of CTCF and consequently TAD fusion occurred in T-ALL (acute lymphoblastic leukemia), resulting in the establishment of spatial interactions between *MYC* and super-enhancer and the up-regulation of *MYC*. In such a process, both CTCF and super-enhancer play vital roles in *MYC* expression. Besides, as mentioned by the reviewer, the genomic distance between enhancer and its targeted gene could be very long (for instance, multi-megabase (Ivelisse et al., Molecular Cell, 2018)), again indicating that the enhancers regulating cluster 3 genes may reside in compartment A and specific TFs tether cluster 3 genes to a euchromatin environment for activation possibly through a phase separation mechanism. By contrast,

unlike cluster 3 genes, as mentioned above, cluster 1 genes tend to locate in an open environment and therefore their regulation may be simpler

The above discussion has been added to the manuscript.

May 9, 2022

RE: Life Science Alliance Manuscript #LSA-2021-01302RR

Prof. Yi Qin Gao
Peking University
Beijing National Laboratory for Molecular Sciences, College of Chemistry and Molecular Engineering
No. 5 Yiheyuan Road
Beijing 100871
China

Dear Dr. Gao,

Thank you for submitting your Research Article entitled "Expression regulation of genes is linked to their CpG density distributions around TSS". It is a pleasure to let you know that your manuscript is now accepted for publication in Life Science Alliance. Congratulations on this interesting work.

DISTRIBUTION OF MATERIALS:

Again, congratulations on a very nice paper. I hope you found the review process to be constructive and are pleased with how the manuscript was handled editorially. We look forward to future exciting submissions from your lab.

Sincerely,
